# Microstructure and Mechanical Properties of Nanoparticulate Y_2_O_3_ Modified AlSi10Mg Alloys Manufactured by Selective Laser Melting

**DOI:** 10.3390/ma16031222

**Published:** 2023-01-31

**Authors:** Fuxu Zhang, Zhenyu Zhang, Qinming Gu, Xuezhang Hou, Fanning Meng, Xuye Zhuang, Li Li, Bingxin Liu, Junyuan Feng

**Affiliations:** 1Key Laboratory for Precision and Non-Traditional Machining Technology of Ministry of Education, School of Mechanical Engineering, Dalian University of Technology, Dalian 116024, China; 2School of Mechanical Engineering, Hangzhou Dianzi University, Hangzhou 310018, China; 3School of Mechanical Engineering, Shandong University of Technology, Zibo 255000, China; 4Yantai Research Institute and Graduate School of Harbin Engineering University, Yantai 264006, China

**Keywords:** selective laser melting, Y_2_O_3_ nanoparticles, AlSi10Mg alloy, microstructure, mechanical properties

## Abstract

AlSi10Mg has a good forming ability and has been widely accepted as an optimal material for selective laser melting (SLM). However, the strength and elongation of unmodified AlSi10Mg are insufficient, which limits its application in the space industry. In this paper, yttrium oxide (Y_2_O_3_) nanoparticles modified AlSi10Mg composites that were manufactured using SLM. The effects of Y_2_O_3_ nanoparticles (0~2 wt.% addition) on the microstructure and mechanical properties of AlSi10Mg alloys were investigated. An ultimate tensile strength of 500.3 MPa, a yield strength of 322.3 MPa, an elongation of 9.7%, a good friction coefficient of 0.43, and a wear rate of (3.40 ± 0.09) ×10^−4^ mm^3^·N^−1^·m^−1^ were obtained with the addition of 0.5 wt.% Y_2_O_3_ nanoparticles, and all these parameters were higher than those of the SLMed AlSi10Mg alloy. The microhardness of the composite with 1.0 wt.% Y_2_O_3_ reached 145.6 HV_0.1_, which is an increase of approximately 22% compared to the unreinforced AlSi10Mg. The improvement of tensile properties can mainly be attributed to Orowan strengthening, fine grain strengthening, and load-bearing strengthening. The results show that adding an appropriate amount of Y_2_O_3_ nanoparticles can significantly improve the properties of the SLMed AlSi10Mg alloy.

## 1. Introduction

Additive manufacturing is a potentially disruptive technology across multiple industries, including automotive, biomedical, and aerospace [1]. The principle of additive manufacturing is to accumulate materials point by point on the surfaces and solidify them into bodies layer by layer [2], enabling the formation of complex parts [3] and increasing product customization [4]. As a dominant AM technology, SLM is highly appreciated for its relatively low surface roughness, high geometric accuracy, and good mechanical properties [5,6]. Based on previously established CAD data, SLM selectively melts a fixed-thickness powder layer on the solidified powder bed, and quickly, layer by layer, produces three-dimensional parts with complex shapes [7,8].

AlSi10Mg belongs to the Al-Si series of aluminum alloys. It has many advantages, such as high specific strength, good thermal conductivity, small thermal expansion coefficient, and excellent casting and welding performance, and it is widely applied in the military, automotive, aerospace, and medical industries [9,10,11]. Although Al-Si alloys have a good forming ability, their strength and elongation are insufficient, which limits their application in the space industry. Therefore, researchers often introduce nanoparticles to obtain a better performance from Al-Si alloys.

Currently, there is much research on preparing AlSi10Mg alloys by adding nanoparticles, such as LaB_6_ [12], CNTs [13,14], GNPs [15], TiC [16], TiB_2_ [17], SiC [18], Al_2_O_3_ [19], AlN [20], BN [21], and TiN [22], to improve mechanical properties. Tan et al. [12] used an edge-to-edge model (E2EM) to study the crystallographic match between Al and other nanoparticles. It was discovered that LaB_6_ is a good candidate as a grain-refining agent, and the LaB_6_-doped AlSi10Mg alloy showed excellent isotropic mechanical properties due to grain refinement and homogeneous organization. Jiang et al. [13] fabricated AlSi10Mg composites containing 1 wt.% CNTs by the ultrasonic stirring of a solution, and the tensile strength reached 499 MPa, which was about 20% higher than without CNTs addition. Tiwari et al. [15] added GNPs to AlSi10Mg powder, and the tensile strength and microhardness of the composites increased by ~20% and ~30%, respectively, by SLM. Gu et al. [16] produced a new ring-mounted TiC at the grain boundaries of the matrix by SLM, and the tensile strength was enhanced without reducing the elongation. However, very little research was reported on the overall improvement of the mechanical properties of aluminum-based nanocomposites, which implies that it is essential to search for a novel nanoparticulate material that can provide an overall enhancement of the mechanical properties.

Rare earth oxides such as Y_2_O_3_ are considered perfect chemical modifiers for Al-Si alloys, since the ratio of atomic radius of Y relative to that of Si is very close to 1.646. The mechanism of silicon modification in aluminum–silicon alloys is impurity-induced twinning [23]. In Moussa’s study, it was discovered that the addition of small amounts of Y_2_O_3_ nanoparticles to Al-Si alloys can reduce the grain size of α-Al and enhance the strength and plasticity of cast samples [24]. Utilizing the refinement and modification mechanism related to the twinning of {111}, the fracture mode gradually transfers from cleavage transgranular fracture to the ductile-brittle fracture mode. Moreover, Y_2_O_3_ was applied to improve the properties of 316L [25], pure tungsten [26], Inconel 625 [27], and other materials processed by SLM, further proving its effectiveness as an oxide dispersion strengthening addition. However, the effect of Y_2_O_3_ nanoparticles on the organization and mechanical properties of AlSi10Mg alloys has not been explored yet. Whether the same grain refinement effect can be achieved for AlSi10Mg is still unknown. Therefore, it is important to investigate the enhancement capability and the strengthening mechanism of Y_2_O_3_ nanoparticles for the AlSi10Mg alloys.

In this study, AlSi10Mg alloys specimens were printed by SLM, adding 0 to 2 wt.% of Y_2_O_3_ nanoparticles. To investigate the effect of the addition of Y_2_O_3_ nanoparticles on the microstructure and mechanical properties, the phase composition, microstructure, grain size, and crystallographic texture of the SLMed specimens were analyzed by XRD, SEM, and EBSD, respectively. Ultimate tensile strength, elongation, yield strength, Vickers hardness, coefficient of friction, and wear rate were also measured.

## 2. Materials and Methods

### 2.1. Materials

A gas-atomized AlSi10Mg powder (Jiangsu Vilory Advanced Materials Technology Co., Ltd., Xuzhou, China, the composition is shown in Table 1) with a particle size ranging from 15 to 53 μm (Figure 1a) and commercial nano-Y_2_O_3_ (Beijing Zhongke Yannuo Advanced Materials Technology Co., Ltd., Beijing, China) was used as raw materials. It should be noticed that the nano Y_2_O_3_ we purchased were in the form of clustered powders as shown in Figure 1b, but the true particle size of Y_2_O_3_ is around 50 nm, as shown in Figure 1c. The A QM-QX2 planetary ball mill (Nanjing Laibu Laboratory Equipment Technology Co., Ltd., Nanjing, China) was used to uniformly disperse Y_2_O_3_ nanoparticles on the surface of the AlSi10Mg powder. The powder mixture was sealed in stainless steel bowls with a ball-to-powder ratio of 3:1. The rotation rate was set at 250 rpm with a total milling time of 4 h. An interval of 10 min was set after each 20 min of milling in order to avoid overheating of powder. The mixed powder was put in a vacuum drying oven, dried at 120 °C for 3 h, and sealed in a vacuum bag for further use. The micro-morphology of the composite powder after ball milling is shown in Figure 2, indicating that the sphericity of the AlSi10Mg powder is still relatively preserved after ball milling. Y_2_O_3_ is easily observed on the powder surfaces. The elemental distribution mapping of Y shows that the Y_2_O_3_ nanoparticles are uniformly distributed on the surface of AlSi10Mg powder after ball milling (Figure 2c–f).

### 2.2. SLM Process

The AlSi10Mg and x wt.%Y_2_O_3_-AlSi10Mg samples were manufactured using an independently developed SLM system (as shown in Figure 3a) using a fiber laser with a power of 500 W, 1064 nm continuous wavelength, and a laser spot diameter of 75 μm (YLR-500 fiber laser, IPG/Germany). To prevent powder oxidation, we performed the SLM process at argon pressure *p* < 0.1 MPa. After the atmosphere was stabilized, the gas flow was controlled at 15 l/min. The optimized SLM parameter combinations were set as follows: laser power (*P*) = 200 W, hatching distance (*H*) = 0.12 mm, laser scanning speed (*V*) = 1200 mm/s, and layer thickness (*L*) = 0.02 mm. The optimized SLM parameter combinations are shown in Table 2. The scan direction angle between neighboring layers was 67°. Cubic specimens with a side length of 10 mm were fabricated for the microstructural analysis. The relative density analysis of the sample was performed using Archimedes’ principle. For simplicity, the SLMed AlSi10Mg samples were named S0, and the composite samples with 0.5, 1.0, 1.5, and 2.0 wt.% nano-Y_2_O_3_ were named S1, S2, S3, and S4, respectively (Table 3).

### 2.3. Microstructure Characterization

The microtopography of the AlSi10Mg/nano-Y_2_O_3_ powders and the microstructure of the SLMed specimens were characterized by an Olympus optical microscopy (OM, BX53M, Olympus, Tokyo, Japan) and a JEOL JSM-IT800SHL scanning electron microscope. The SLMed specimens were pre-treated using mechanical grinding, polished using standard metallographic methods, and then etched by Keller’s reagent (1 mL HF, 1.5 mL HCl, 2.5 mL HNO_3_, and 95 mL H_2_O). The phase composition of the specimens was analyzed using PANalytical X-ray diffraction with a step size of 0.02° and the 2θ range from 20 to 90°. The grain size and texture of the composites were analyzed by electron backscatter diffraction (EBSD).

### 2.4. Mechanical Tests

The printing direction of all specimens are shown in Figure 3b. All specimens are removed from the substrate by Wire Electrical Discharge Machining. Before mechanical tests, all specimens were pre-treated using mechanically grinding and polished using standard metallographic methods. According to the ASTM E8 standard, specimens for the mechanical tensile test were machined into flat tensile coupons with a pitch length of 25 mm, a width of 6 mm, and a thickness of 2 mm (Figure 4). Room-temperature tensile tests were operated on a tensile testing machine (C45, MTS, Minnesota, USA) with a constant extension rate of 1 mm/min at room temperature. To ensure accurate measurement of strain in tensile tests, an extensometer was used. At least three samples for each parameter were printed and all tests were conducted three times, and the average value was taken as the final result. The Vickers hardness (HV) of the samples was measured with an HVS-1000A microhardness tester (Laizhou Huayin Test Instrument Co., Ltd., Yantai, China) at room temperature with a load of 0.98 N and a duration time of 15 s. The wear/tribological properties of the specimens were evaluated by a CF-I reciprocating tribometer (Lanzhou ZhongKe KaiHua Sci. and Technol. Co., Ltd., Lanzhou, China). The tribometer was equipped with a silicon nitride ball with a diameter of 6 mm. The friction unit was slid at a speed of 0.05 m/s for 30 min under 5 N load. The track of the friction pair was 5 mm. Each specimen was first polished and then conducted for three different runs. The coefficient of friction (COF) was recorded during the tests. The wear volume (V) of different parts was calculated by a MT-500 probe type abrasion mark measuring instrument (Lanzhou ZhongKe KaiHua Sci. and Technol. Co., Ltd., Lanzhou, China). The wear rate (ω) was identified by ω = V/WL, where W was the contact load applied in the test and L was the sliding distance.

## 3. Results and Discussion

### 3.1. Phase Composition and Microstructural Characterization

Figure 5 shows the X-ray diffraction patterns (XRD) of the SLMed AlSi10Mg and Y_2_O_3_-AlSi10Mg specimens. All of the samples have similar diffraction peaks, containing the main peaks of the face-centered cubic (FCC) α-Al and eutectic Si phases. Si is supersaturated and precipitates in the Al matrix because of the rapid cooling during the SLM process. In addition, the Al_3_Y phase can be found in SLMed Y_2_O_3_-AlSi10Mg samples. The diffraction peaks of (111) and (200) of Si are similar in intensity, which can be observed in Figure 4a. The other samples have a preferred orientation, and the strongest diffraction peak for all of them (200). The diffraction peaks are locally amplified to analyze the variation in the α-Al matrix phase (Figure 4b). The careful comparison indicates that the diffraction peak corresponding to the α-Al in the SLMed Y_2_O_3_-AlSi10Mg specimens has a slight shift to the left compared with the SLMed AlSi10Mg, and the shift increases with the nano-Y_2_O_3_ content. According to the following Bragg formula in Equation (1) and the crystallographic relationship in Equation (2), the following is the case:(1)$$2\;d_{h\;k\;l}\sin\theta =\lambda$$
(2)$$d_{h\;k\;l}=\frac{a}{\sqrt{h^{2}+k^{2}+l^{2}}}$$
where *θ* denotes the diffraction angle, *d*_hkl_ represents the crystal plane spacing, *λ* is the X-ray wavelength, h, k, and l are the Miller indices of the crystal, and a is the lattice constant. It can be inferred that when the diffraction peak shifts to the left, the lattice constant becomes larger.

Figure 6 shows the typical microstructure of SLM samples. As shown in Figure 6a–e, the SLMed samples have overlapping fish-scale molten pools. The molten pool of the SLM samples is randomly distributed and their size is uneven. Under the same process parameters, the relative density first increases and then decreases with the yttrium oxide content, as shown in Table 3. The maximum relative density is 99.45% when 0.5 wt.% Y_2_O_3_ nanoparticles are added. The SEM image of the longitudinal section further visualizes the details of the microstructure of the SLM samples, as shown in Figure 7. The boundaries of the molten pool trajectory are visible. The alloy’s microstructure consists of a cell-like Al matrix (dark phase) and a fibrous eutectic silicon grid (light phase). Thus, the main reason for the occurrence of fibrous Si is the rapid solidification during SLM [14]. Inside the molten pool, cell-like dendrites grow toward the center of the molten pool. Two zones can be distinguished according to their morphology and size: a fine cellular zone at the center of the melt pool and a coarse cellular zone at the molten pool boundary [3]. The cells in samples modified by nano-Y_2_O_3_ are relatively refined (especially cells at the center of the molten pool) compared to the SLMed samples.

The cooling and solidification mode of the molten pool mainly depends on the time when the powder is in contact with the laser, and the temperature gradient G and the crystal growth rate R are the two factors affecting the grain growth morphology in the molten pool [8,28]. The G is the temperature difference over a certain distance dT/dx and varies over the time and place inside the melt pool. The R depends on the laser scanning speed and the angle between the laser moving direction and the growth direction of the solidifying material [29]. The G/R ratio determines the microstructural morphology. G × R represents the cooling rate, which determines the fineness of the grains, and the larger the product, the finer the microstructure. According to the Gaussian distribution of laser energy, the degree of undercooling gradually decreases along the cross-section, with a maximum at the center of the molten pool and a minimum at the boundary of the molten pool [30]. Therefore, two distinct areas can be seen in the microstructure of the SLMed specimens.

### 3.2. Grain Size and Crystallographic Texture

To further investigate the microstructure characteristics, S0, S1, and S4 samples are treated with EBSD as shown in Figure 8, which reveals a significant difference between the SLMed Y_2_O_3_-AlSi10Mg specimens with different Y_2_O_3_ contents. Using IPF (inverse pole figure) as a reference, it can be found that most of the grains in Figure 9a–c have the crystal orientations of (001), (101), and (111). As shown in Figure 8a, the grain morphology of specimen S0 exhibits obvious heterogeneity. In addition, the coarse columnar grains are almost parallel to the building direction. The maximal intensity of the pole figures of the S0 reaches 4.19, as revealed in Figure 9a, which indicates that the SLMed AlSi10Mg alloy has a strong texture. After adding 0.5 wt.% Y_2_O_3_ nanoparticles, many small grains nucleated. The IPF color, as shown in Figure 9c, indicates that the equiaxed grains of S1 are randomly oriented. The maximal intensity of the S1 pole figures is only 1.55, which indicates that S1 has almost no texture, as shown in Figure 9b. The molten pool boundary inside S1 is difficult to distinguish due to the high homogeneity of grain morphology. Figure 9b shows that S4 has fine equiaxed crystals, small columnar crystals, and coarse equiaxed crystals. Meanwhile, the highest intensity of the S4 pole figures is 3.25.

The effect of the Y_2_O_3_ content on the grain size of the Y_2_O_3_-AlSi10Mg composite specimens is investigated further. The detailed exploration is based on the EBSD orientation mappings, and the resulting statistics are shown in Figure 10a,c,e. The average grain size of the SLMed Y_2_O_3_-AlSi10Mg specimens with different Y_2_O_3_ contents (0 wt.%, 0.5 wt.%, and 2.0 wt.%) is 1.742, 0.873, and 1.118 μm. Grain boundary angle misorientation is also an important indicator of the microstructural characteristics of materials [22]. The grain boundary angles can be segmented into high-angle grain boundaries (HAGBs > 15°) and low-angle grain boundaries (LAGBs < 15°). Figure 10b,d,f, shows the grain boundary misorientation distribution of S0, S1, and S4, respectively. The volume fractions of the HAGBs in S0, S1, and S4 are 34.4, 64.9, and 54.8%, respectively. In the framework of classical dislocation models, LAGBs can be described in terms of dislocation density [31]. The volume fraction of the LAGBs is proportional to the dislocation density.

The grain refinement ability of yttrium oxide is also verified by the edge-to-edge model (E2EM). The E2EM established by Zhang and Kelly was originally used to test the actual atomic matching at the interface between two phases, and in recent years, it has been used to find a grain refining agent that promotes the heterogeneous nucleation of grains in alloys [32,33,34]. The model is based on the assumption that the orientation relationship of any two phases is determined by the minimum interfacial strain energy, and the two phases are atomically matched to minimize the strain energy. It calculates the orientation relationships between two phases based on the lattice constant, crystal structure, and atomic position data. To obtain the best matching relationship, the atom matching direction must be carried out along the dense and sub-dense crystal directions, and the interatomic spacing misfit (*f_r_*) must be less than 10%. These crystal directions are called matching directions, and two pairs of matching crystal directions can uniquely determine a pair of crystal faces, and if the interplanar spacing mismatch (*f_d_*) is also less than 10%, the particle can serve as a heterogeneous nucleation core of the matrix [12].

Al has a lattice constant of 0.405 nm and is a face-centered cubic (fcc) structure [28]. It contains three possible close-packed or nearly close-packed directions: <110>_Al_, <100>_Al,_ and <112>_Al_, and three close and nearly close-packed planes: {111} _Al_, {200} _Al,_ and {220} _Al_. Y_2_O_3_ has a cubic structure with a lattice parameter of 1.06 nm. There are three potential close-packed directions: <110>_Y2O3_, <100> _Y2O3,_ and<112> _Y2O3_. {222} _Y2O3_, {400} _Y2O3,_ and {440} _Y2O3_ are the close and nearly close-packed planes for Y_2_O_3._ There are two direction pairs, <100>_Al_//<110> _Y2O3_ and <112>_Al_//<100> _Y2O3_, with the *f_r_* value of 7.46% and 6.85%, respectively, which are both less than 10%. Further, the valid close-packed plane pairs include {200}_Al_//{044}_Y2O3,_ with an *f_d_* value of 7.47%. According to the E2EM crystallographic geometric model, the final predictive orientation relationship between Y_2_O_3_ and Al matrix is as follows:[100]_Al_//[110]_Y2O3_, (200)_Al_//(044)_Y2O3_

According to the E2EM model, the interplanar spacing mismatch and the interatomic spacing misfit of the orientation relationships between Y_2_O_3_ and Al matrix are presented in Figure 11. The calculated values are less than 10%. Therefore, the Y_2_O_3_ nanoparticles can be used as effective grain refiners. A large number of Y_2_O_3_ nanoparticles can supply substantial heterogeneous nucleation sites for α-Al, which can effectively facilitate the nucleation of α-Al grains. As shown in Figure 8, the EBSD images illustrate the significant refinement of alloy grains doped with Y_2_O_3_ nanoparticles compared to undoped AlSi10Mg alloys.

### 3.3. Mechanical Properties

Figure 12 depicts the tensile properties of the SLM-processed AlSi10Mg and Y_2_O_3_-AlSi10Mg alloys at room temperature. The data of the ultimate tensile strength (UTS), elongation to failure (El), yield strength (YS), and Vickers hardness (HV_0.1_) are listed in Table 4. The SLM-processed AlSi10Mg alloy has an ultimate tensile strength of 431.2 MPa, yield strength of 264.4 MPa, and an elongation of 7%. After adding 0.5 wt.% of nano-Y_2_O_3_ particles, the S1 sample exhibits an ultimate tensile strength of 500.3 MPa, an elongation of 9.7%, and a yield strength of 322.3 Mpa. In general, using optimal processing parameters, the SLM-fabricated Y_2_O_3_-AlSi10Mg specimen exhibits considerably higher strength than the unreinforced AlSi10Mg alloy specimen. The UTS of S1 is increased by 16% compared to S0, and the yield strength and elongation are also improved. With the increase in the nanoparticles content, the microhardness first increases and then decreases, as shown in Table 4. When 1.0 wt.% Y_2_O_3_ nanoparticles is added, the microhardness is up to 145.6 HV_0.1_, which is 21.8% higher than the 119.5 HV_0.1_ of S0.

The S1 sample exhibits great ultimate tensile strength (500.3 MPa), which is 69.1 MPa higher than that of the S0 sample, as shown in Table 4. The high strength of SLMed Y_2_O_3_-AlSi10Mg samples can mainly owe to Orowan strengthening, fine grain strengthening, and load-bearing strengthening.

The Orowan strengthening, caused by the resistance of tightly packed hard particles to the dislocation passing, is a very important item in the aluminum alloy strengthening mechanism [35]. For reinforced particles with an average diameter of 5 μm or larger, that mechanism is not a major factor. In contrast, the Orowan strengthening is more pronounced when highly dispersed nanoparticles are present in the aluminum matrix [36]. It has been established that insoluble nanoparticles in the matrix can significantly improve the ability to hinder dislocations, even for only a small volume fraction. For composites containing nanoparticles, this is usually explained by the Orowan strengthening mechanism:(3)$$\Delta \sigma_{Orowan}=\frac{2G_{m}b}{d_{p}}{\left(\frac{6V_{p}}{\pi }\right)}^{1/3}$$
where d_p_ and V_p_ are the size of the Y_2_O_3_ nanoparticles (~50 nm) and the volume fraction (~0.938 vol%) of S1, respectively. The term b is the Burgers vector (~0.286 nm), and G_m_ is the shear modulus of the matrix (~26.5 GPa for Al). The result of the calculated Δσ_Orowan_ is 79.3 MPa.

We all know that the grain boundary strength is stronger than that of grains [37]. The finer the Al matrix grain, the more the grain boundaries, impeding dislocation motion. According to the Hall–Petch formula [38], when the grain size decreases and the grain boundary increases, the strength of the material increases.
(4)$$\sigma_{Hall-Petch}=\sigma_{0}+kd^{-\frac{1}{2}}$$
where *d* denotes the average grain size of metal, and *k* means the material constant. According to previous studies, the *k* for the Al alloy is set to 50 MPa μm^1/2^ [39]. As shown in Figure 9, the average grain sizes of S0 and S1 are 1.742 and 0.873 μm. The calculated *Δσ_Hall-Petch_* is 15.6 MPa.

The contribution of the load-bearing strengthening to the strength of the nanocomposites (Δ*σ_Load_*) can be expressed as follows [40]:(5)$$\Delta \sigma_{Load}=1.5V_{p}\sigma_{m}$$
where *σ_m_* is the YS of the AlSi10Mg alloy matrix (264.4 MPa). V_p_ is the volume fraction of the nano-Y_2_O_3_. The result of the calculated Δ*σ_Load_* is 3.72 MPa. Therefore, the enhancement in tensile strength in this study can be illustrated as follows:(6)$$\Delta \sigma =\Delta \sigma_{\mathrm{Hall}-\mathrm{Patch}}+\Delta \sigma_{\mathrm{Orowan}}+\Delta \sigma_{\mathrm{Load}}$$

The calculated total theoretical increment of the YS (Δ*σ* = 98.62 MPa) is higher than the actual yield strength enhancement (67.9 MPa) due to the presence of holes, and the calculated theoretical increment does not reflect the actual enhancement of nano-Y_2_O_3_. It is worth noting that the tensile strength does not increase with the addition of Y_2_O_3_ nanoparticles. The effect of relative density of SLMed samples on the mechanical and microstructural properties of the final produced component (e.g., tensile strength, defects formation, fracture toughness, friction and wear) is undeniable [41,42,43]. The elimination of pores is necessary for obtaining simultaneously enhanced strength and ductility. Therefore, the reason for the decrease in the strength increment of S2, S3, and S4 is that the porosity increases with respect to the nano-Y_2_O_3_ content.

To better comprehend the fracture mechanism of the SLMed Y_2_O_3_-AlSi10Mg specimens, the fracture morphology of S0, S1, S2, S3, and S4 are analyzed by SEM. As shown in Figure 13a–e, all samples can observe the cleavage planes and dimples from the fractography, revealing a mixed fracture pattern of ductile and brittle fracture. The high-magnification SEM of the fracture morphology is shown in Figure 13f–j, and it can be observed that there are some ligament fossae with a size less than 0.5 μm on the fracture surface, and that the shape of the dimples is regular, but the depth is different. Deep and dense dimples indicate the high elongation of S1, as shown in Figure 13g. As the Y_2_O_3_ content increases, some holes appear, which leads to an elongation decrease.

Figure 14 shows the coefficient of friction (COF) and the corresponding wear rate of the AlSi10Mg and Y_2_O_3_-AlSi10Mg alloys. The COFs of Y_2_O_3_-AlSi10Mg composites are higher than those of AlSi10Mg specimens in the initial stage, but the COFs are lower than that of AlSi10Mg specimens in the steady state. The COF and wear rate of the AlSi10Mg are 0.68 and (5.59 ± 0.15) × 10^−4^ mm^3^·N^−1^·m^−1^, concurrently, and the COF curve exhibits a certain fluctuation. With the addition of Y_2_O_3_ nanoparticles, the wear first decreases and then increases. When adding 0.5 wt.% Y_2_O_3_ nanoparticles, the COF of the composites decreases from 0.68 to 0.43, and the wear rate is also significantly reduced, from (5.59 ± 0.15) × 10^−4^ mm^3^·N^−1^·m^−1^ to (3.40 ± 0.09) × 10^−4^ mm^3^·N^−1^·m^−1^. The COF curve of Y_2_O_3_-AlSi10Mg alloys is relatively stable.

To further investigate the friction and wear mechanism, the wear surface topography was inspected and is shown in Figure 15a–e. For the S0, the worn surface showed deep grooves and debris (Figure 15a). Therefore, a combined mechanism of abrasive and adhesive wear dominated, showing a relatively high average COF of ~0.709 and a high wear rate of (5.59 ± 0.15) × 10^−4^ mm^3^·N^−1^·m^−1^ for S0. For the S1 and S2, small delamination and pits are observed on the smooth worn surface, revealing a weak wear behavior in this case, and implying that the dominant wear mechanism has changed to the adhesion of a strain-hardened tribolayer. Owing to their high hardness, a high-density of hard reinforcements dispersed in the soft matrix can effectively limit further material removal in the course of the wear process [44]. However, the COF of the S4 increased to 0.65, and the resultant wear rate also increased slightly to (4.67 ± 0.18) × 10^−4^ mm^3^·N^−1^·m^−1^. The decrease in tribological performance in this instance was ascribed to the decrease in density and the grain coarsening (Figure 8b), which will easily cause material delamination during sliding, resulting in the spalling of the worn surface and attendant-limited wear resistance.

## 4. Conclusions

The AlSi10Mg composites modified with Y_2_O_3_ nanoparticles were prepared by SLM. The effect of nano-Y_2_O_3_ particles (0−2 wt.% addition) on the microstructure and mechanical properties of AlSi10Mg alloy was studied under the same process parameters. The main conclusions are as follows:Adding 0.5 wt.% Y_2_O_3_ nanoparticles can significantly refine the grains from 1.742 to 0.873 μm, but further addition of Y_2_O_3_ nanoparticles will result in a grain size increase and the decrease in the relative density;The optimal nano-Y_2_O_3_ particle addition level is 0.5 wt.%, with a higher ultimate tensile strength of 500.3 MPa, a yield strength of 322.3 MPa, and an elongation of 9.7%. Both the Orowan strengthening effect and the load-bearing strengthening effect show that the addition of nano-Y_2_O_3_ is beneficial to the grain refinement. However, since the grain size gradually increases and the relative density decreases as the addition level of Y_2_O_3_ passes 1 wt%, the strength of the material also experiences a decrease;The wear resistance of the Y_2_O_3_-AlSi10Mg nanocomposites is improved compared to that of the AlSi10Mg alloy. When adding 0.5 wt.% Y_2_O_3_ nanoparticles, the wear rate is about 39% lower than that of the AlSi10Mg alloys, but when the addition of Y_2_O_3_ increases, the wear performance gradually decreases.

## Figures and Tables

**Figure 1 materials-16-01222-f001:**
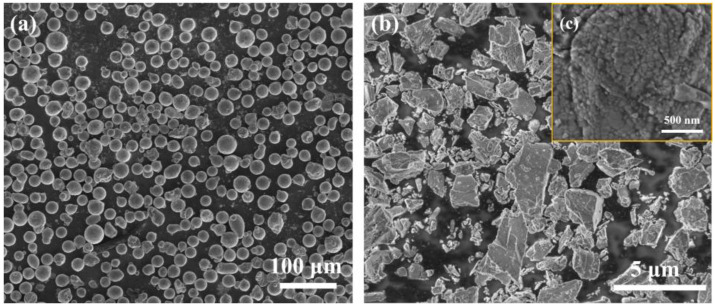
SEM images of (**a**) AlSi10Mg powder, (**b**) Y_2_O_3_ clustered powder, and (**c**) nano Y_2_O_3_ particles that form the clustered powder.

**Figure 2 materials-16-01222-f002:**
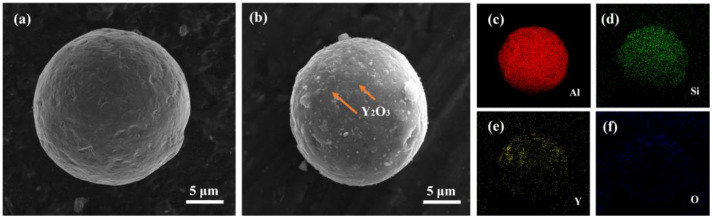
SEM image showing the morphology of (**a**) AlSi10Mg power and (**b**) 0.5 wt.% Y_2_O_3_-AlSi10Mg power. The orange arrows point to the Y_2_O_3_ nanoparticles. EDS mapping showing the elemental distribution of (**c**) Al, (**d**) Si, (**e**) Y, and (**f**) O on the particle surface.

**Figure 3 materials-16-01222-f003:**
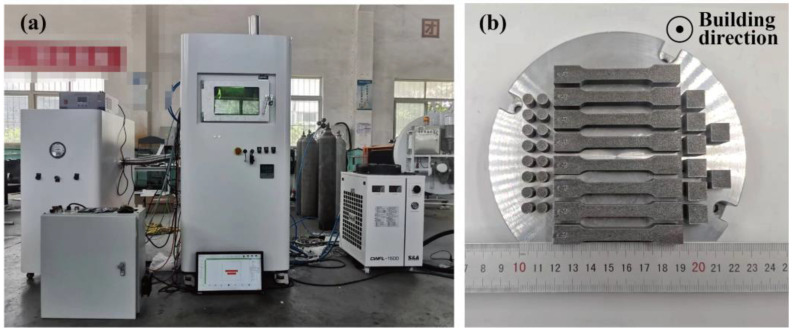
(**a**) The photograph of SLM printer, (**b**) the samples printed on AlSi10Mg substrate.

**Figure 4 materials-16-01222-f004:**
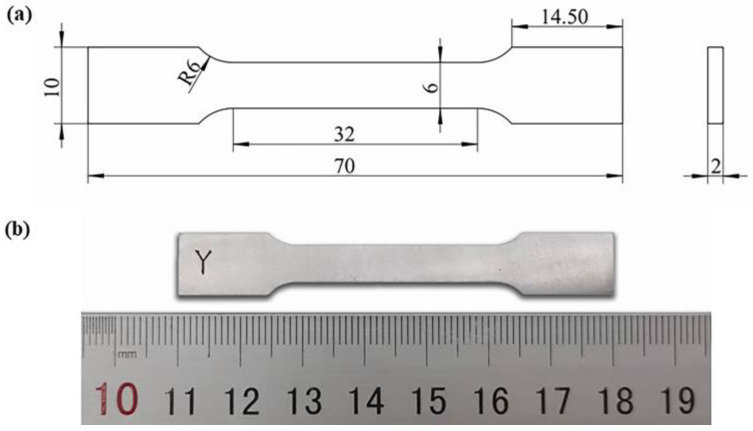
(**a**) Dimensions of the tensile test sample (mm) and (**b**) the image of the polished coupon.

**Figure 5 materials-16-01222-f005:**
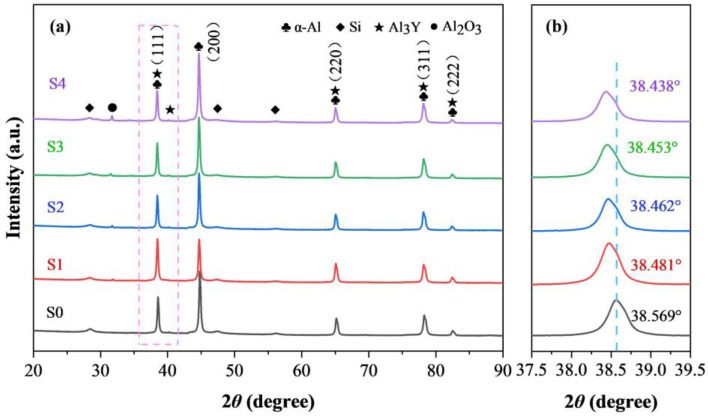
XRD patterns of (**a**) all samples and (**b**) magnification of the selected local areas in (**a**).

**Figure 6 materials-16-01222-f006:**
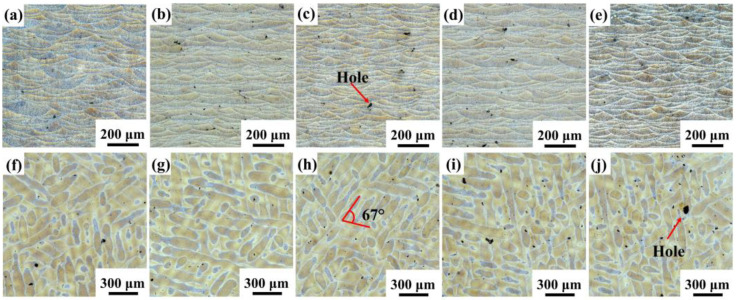
Optical microscopy images of S0, S1, S2, S3, and S4 samples in the (**a**–**e**) longitudinal section and (**f**–**j**) cross section; (**a**,**f**) S0; (**b**,**g**) S1; (**c**,**h**) S2; (**d**,**i**) S3; (**e**,**j**) S4.

**Figure 7 materials-16-01222-f007:**
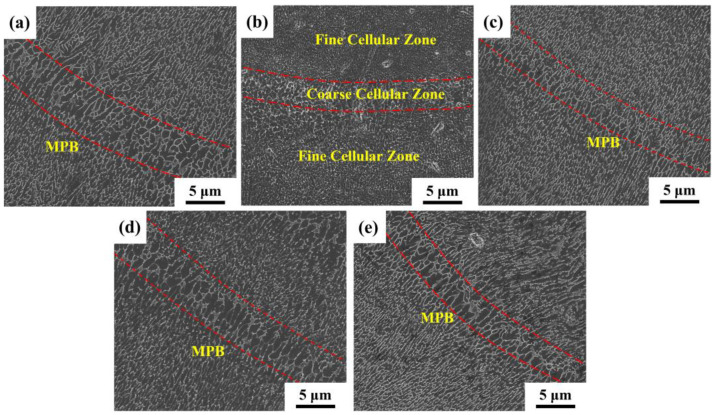
SEM images of SLMed samples: (**a**) S0; (**b**) S1; (**c**) S2; (**d**) S3; (**e**) S4.

**Figure 8 materials-16-01222-f008:**
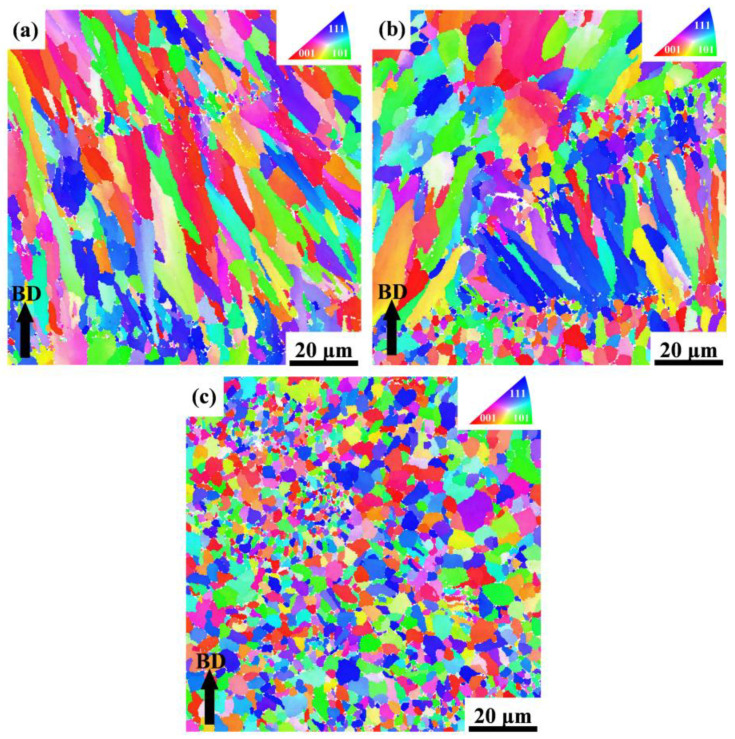
EBSD orientation maps of the Al grains across their building direction in (**a**) S0, (**b**) S4, and (**c**) S1. The IPF color code represents the grain orientation.

**Figure 9 materials-16-01222-f009:**
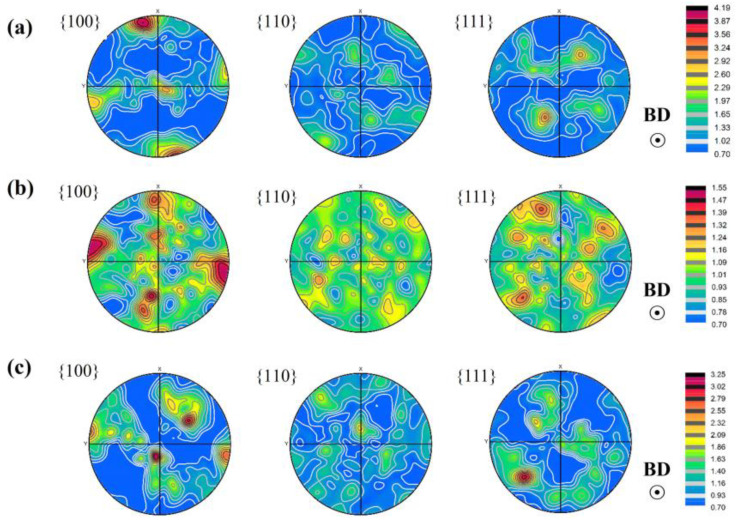
Pole figures of the longitudinal direction: (**a**) S0, (**b**) S1, (**c**) S4.

**Figure 10 materials-16-01222-f010:**
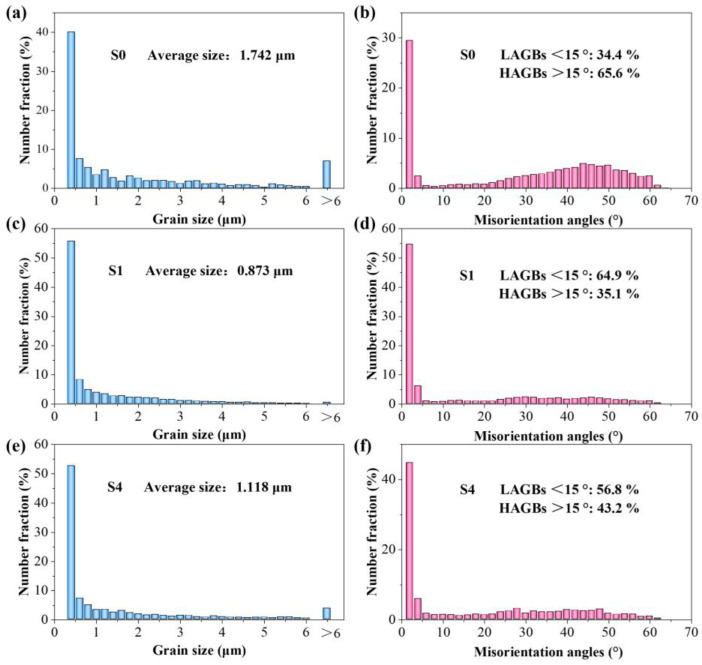
Grain size distribution statistics (**a**,**c**,**e**) and grain boundary misorientation angles distribution statistics (**b**,**d**,**f**); (**a**,**b**) S0; (**c**,**d**) S1; (**e**,**f**) S4.

**Figure 11 materials-16-01222-f011:**
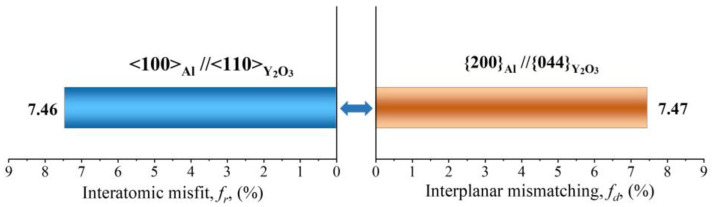
Schematic diagram of the interplanar spacing mismatch and the interatomic spacing misfit between Y_2_O_3_ and Al.

**Figure 12 materials-16-01222-f012:**
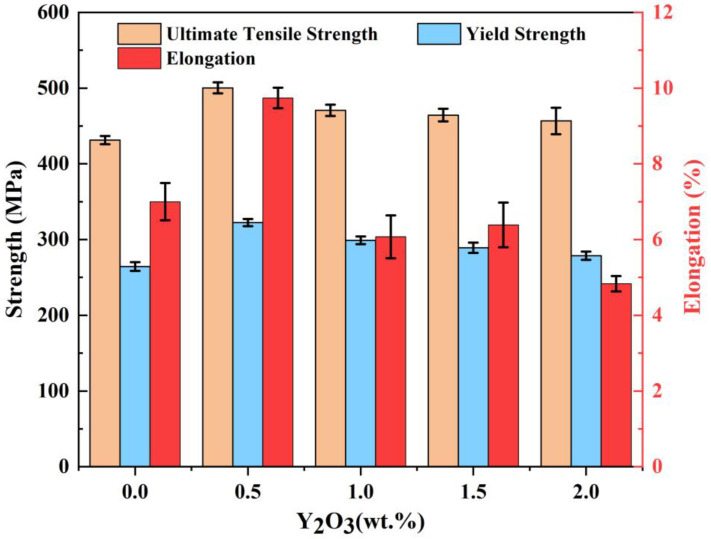
Ultimate tensile strength, yield strength, and elongation data of the SLMed AlSi10Mg and Y_2_O_3_-AlSi10Mg alloy.

**Figure 13 materials-16-01222-f013:**
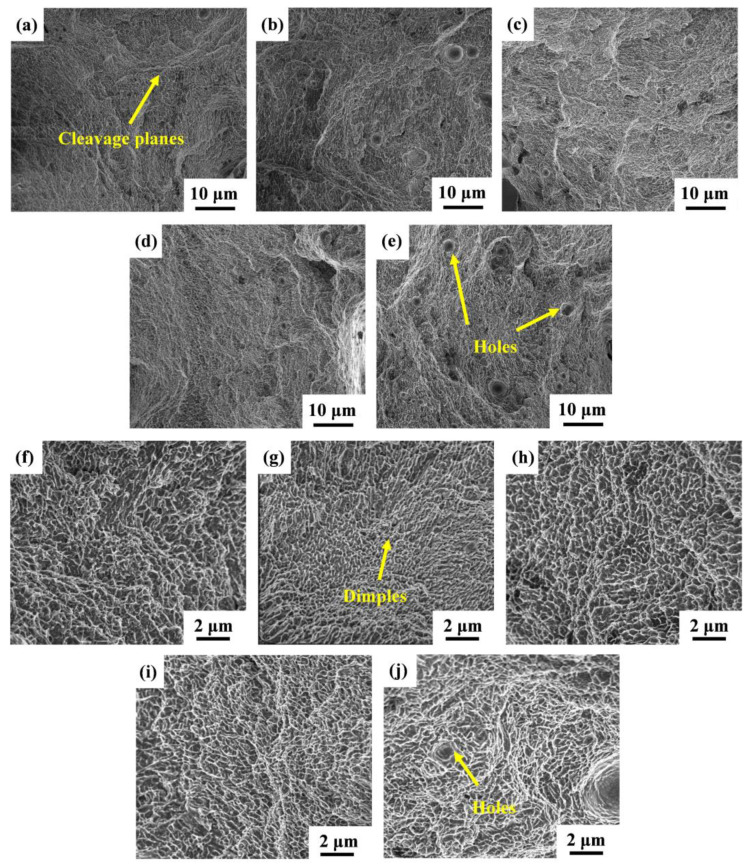
SEM images of fracture surfaces: (**a**,**f**) for S0; (**b**,**g**) for S1; (**c**,**h**) for S2; (**d**,**i**) for S3; (**e**,**j**) for S4.

**Figure 14 materials-16-01222-f014:**
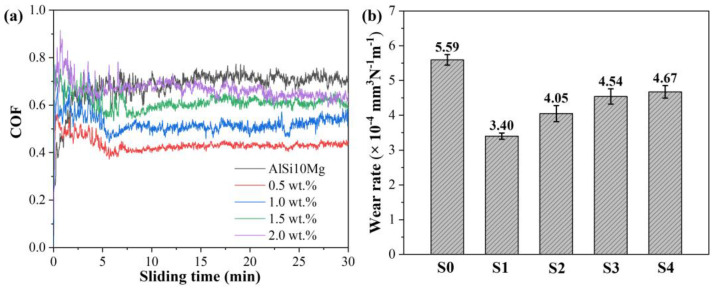
Wear properties of the SLMed part: Coefficient of friction (**a**) and the wear rate (**b**).

**Figure 15 materials-16-01222-f015:**
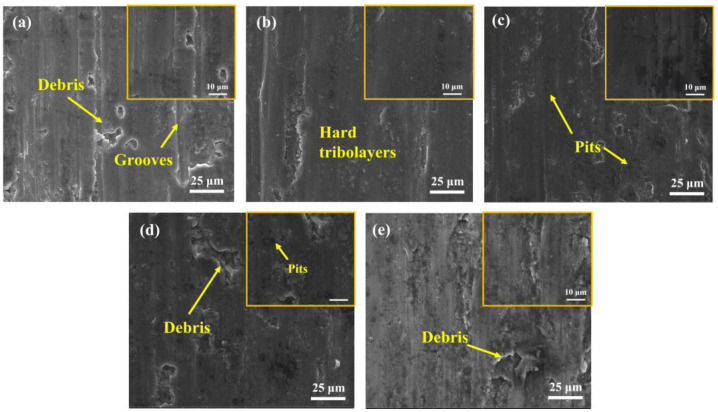
Wear surface morphologies of (**a**) S0; (**b**) S1; (**c**) S2; (**d**) S3; (**e**) S4.

**Table 1 materials-16-01222-t001:** Chemical composition of the AlSi10Mg powder (wt.%).

Al	Si	Mg	Fe	Zn	Mn	Cu	Ni	Ti	Pb	Sn
Bal.	10.23	0.33	0.073	0.011	<0.005	<0.005	<0.005	<0.005	<0.005	<0.005

**Table 2 materials-16-01222-t002:** The optimized SLM parameter combinations.

Laser Power (W)	Hatching Distance (mm)	Scanning Speed (mm/s)	Layer Thickness (mm)
200	0.12	1200	0.02

**Table 3 materials-16-01222-t003:** Composition of the mixture powder (wt.%) and relative density (%) of different samples.

Samples	AlSi10Mg Content	Y_2_O_3_ Content	Relative Density
S0	100	0	98.89
S1	99.5	0.5	99.45
S2	99.0	1.0	99.20
S3	98.5	1.5	98.62
S4	98.0	2.0	98.13

**Table 4 materials-16-01222-t004:** Mechanical properties of SLM-processed specimens.

Specimens	Ultimate Tensile Strength (MPa)	Yield Strength (MPa)	Elongation (%)	Vickers Hardness (HV_0.1_)
S0	431.2 ± 5.4	264.4 ± 5.8	7 ± 0.5	119.5 ± 3.5
S1	500.3 ± 7.1	322.3 ± 4.7	9.7 ± 0.3	128.8 ± 5.1
S2	470.8 ± 7.4	298.9 ± 5.0	6.1 ± 0.6	145.6 ± 5.9
S3	464.3 ± 8.4	289.2 ± 6.8	6.4 ± 0.6	128.4 ± 3.9
S4	456.6 ± 17.6	273.8 ± 5.4	4.8 ± 0.2	129.2 ± 5.3

## Data Availability

Not applicable.

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
