# Peer review of "Microstructure and Mechanical Properties of Nanoparticulate Y2O3 Modified AlSi10Mg Alloys Manufactured by Selective Laser Melting"

_materials, 2023, doi:10.3390/ma16031222_

Round 1
Reviewer 1 Report
Dear Authors
I have some suggestions and questions to Your manuscript. Here are some of them. The minor corrections are highlighted in Yellow in the attached file.
1. Line 86. You say, that Y2O3 particles were 50 nm, but on Figure 1b they are 3-6 um. They are not nanoparticles.
2. Figure 2. The arrowed particles are rather fragments of the alloy powder (See Fig. 1a, inside the red loops), but not Y2O3. The EDS also says, that they are not Y2O3. Please, provide here reliable proof of Your words. Please, also, provide the micrograph of pure AlSi10Mg particle with the same size and magnification (as on Fig. 2a), side-by-side to compare.
3. Line 106-107. To prevent powder oxidation, we performed the SLM process in an argon atmos phere of less than 0.05 vol.%... Please, describe the shielding atmosphere more accurately. 0.05% of what? What was the pressure of the gas, and gas flow (l/min)?
4. Please, rearrange the text for the tables were entirely on the single page, and not as Tab 1 and Tab 2.
5. Line 139-141. Please, specify the test conditions. Usually, reciprocating tribometer can not rotate. It may slide with some linear speed.
6. Figure 4. Please, explain, why the Al2O3 was not identified on XRD pattern.
7. Figure 5. Please, rearrange the Figure. Place the pictures in the correct order.
8. For Figure 6, please, use bigger magnification. The Y2O3 nanoparticles are not visible there.
9. Line 202-203. Please, explain, why You did not do the EBSD for the samples S2 and S3.
10. Lines 207-209. You describe not the molten pool, but the section of rapidly solidified metal. They are very different, please, correct the explanation.
11. Figure 14 is described very inaccurately. The 4 images contain abrasive grooves and delamination, but in different ratios. Fig. 4bd have especially intensive delamination. And, I disagree, that on Fig. 14d in the oval there are wear debris. It is a site of delamination. For more careful analyses, the other, higher magnification images should be made. All 5 alloys should be analyzed. The pores on Fig. 14d are close in shape and size to the Y2O3 “nanoparticles” from Figure 1b.
12. Figure 13 should be discussed more broadly. Why the COF rapidly decreases, when Y2O3 content is 0.5%, and then increases? What is the reason? High COF is not a result of deep grooves. It is a reason, why the grooves become deeper. Please, find the actual reason of changing the COF. For this, as I recommend, use the higher magnification images (X2000, X5000).
13. After the article is improved, please, rewrite the conclusions taking into account new results obtained.

Author Response
Dear reviewer,
Thank you very much for your review comment. We have revied the manuscript according to your comments. Shown below is the detailed information on how I responded to the comments, highlighting the revisions made in the manuscript. When reading the information below, please note:
Text in italic style: the comments
Text in regular style: my responses to the comments.
Changes are highlighted in red color in the revised manuscript.
REVIEWER REPORT(S):
Comments to the Author
Recommendation: Publish after major revisions noted.
I have some suggestions and questions to Your manuscript. Here are some of them. The minor corrections are highlighted in Yellow in the attached file.
Thank you for your corrections, we have revised the whole manuscript accordingly.
Questions:
- Line 86. You say, that Y2O3 particles were 50 nm, but on Figure 1b they are 3-6 um. They are not nanoparticles;
Thank you for pointing that out. The Figure we presented was indeed misleading. We have already added an enlarged image of the nano particles. Nanoparticles are very prone to agglomeration, therefore, the Y2O3 we purchased was in the form of large chunks as large as 3-6 μm. But as can be seen on the newly added image, the actual size of the particles are in the range of dozens of nanometers. Through ball milling, the nanoparticles can be evenly distributed with AlSi10Mg particles without affecting subsequent experiments.
- Figure 2. The arrowed particles are rather fragments of the alloy powder (See Fig. 1a, inside the red loops), but not Y2O3. The EDS also says, that they are not Y2O3. Please, provide here reliable proof of Your words. Please, also, provide the micrograph of pure AlSi10Mg particle with the same size and magnification (as on Fig. 2a), side-by-side to compare.
We are sorry for creating such confusion. According to your comment, we have provided the micrograph of pure AlSi10Mg particle with the same size and magnification (as on Fig. 2a). It can be seen from the micrograph that pure AlSi10Mg particle is very smooth, even though there are small fragments around. The initial thought was to illustrate the distribution of Y2O3 by showing only the distribution of the Y element, so the EDS mapping of the O element were not shown in the paper. From the EDS mappings, it can be seen that the distribution of the Y element and the O element almost coincide. The elemental distribution mappings of Y and O show that the Y2O3nanoparticles are uniformly distributed on the surface of AlSi10Mg powder after ball milling. Similar EDS result was also used to identify the Y2O3 addition in other works as shown in the following figure.
[1] Hu, Zhangping, et al. "Simultaneous enhancement of strength and ductility in selective laser melting manufactured 316L alloy by employing Y2O3 coated spherical powder as precursor." Journal of Alloys and Compounds 899 (2022): 163262.
- Line 106-107. To prevent powder oxidation, we performed the SLM process in an argon atmosphere of less than 0.05 vol.%... Please, describe the shielding atmosphere more accurately. 0.05% of what? What was the pressure of the gas, and gas flow (l/min)?
Thank you, we have added the detailed description for the shielding atmosphere.
To prevent powder oxidation, we performed the SLM process in an argon atmosphere of less than 0.1 MPa. After the atmosphere is stabilized, the gas flow is controlled at 15 l/min.
- Please, rearrange the text for the tables were entirely on the single page, and not as Tab 1 and Tab 2.
Thank you for pointing that out. We have adjusted the position of all tables.
- Line 139-141. Please, specify the test conditions. Usually, reciprocating tribometer can not rotate. It may slide with some linear speed.
Thank you for your comment. The wear/tribological properties of the specimens were evaluated by a CF-I reciprocating tribometer (Lanzhou ZhongKe KaiHua Sci. & Technol. Co., Ltd., China). The friction unit was slid at a speed of 0.05 m/s for 30 min under 5 N load. The track of the friction pair was 5 mm. The calculated linear speed is 0.05 m/s. We have supplemented the linear speed to the test conditions.
- Figure 4. Please, explain, why the Al2O3 was not identified on XRD pattern.
The Al2O3 was identified on the XRD pattern as shown in the red rectangular region.
- Figure 5. Please, rearrange the Figure. Place the pictures in the correct order.
Thank you very much for pointing out that Figure 5 was not properly arranged. We have already rearranged Figure 5.
- For Figure 6, please, use bigger magnification. The Y2O3 nanoparticles are not visible there.
Thank you for pointing that out. The main purpose of Figure 6 is to show the microstructure of the alloys, not to prove the presence of Y2O3 nanoparticles. We actually took microscopic pictures at bigger magnification (X40000, X80000), but the Y2O3 nanoparticles were not identical since they were too small and too sparsely distributed.
- Line 202-203. Please, explain, why You did not do the EBSD for the samples S2 and S3.
We have conducted EBSD for undoped S0 for comparison, for S1 since it has the best tensile strength, and for S4 since it has the highest proportion of Y2O3 nanoparticles doped. The comparison S1 and S4 is to explain the effect of different doping content on grain size and grain boundary angle. We considered that the analysis for these three samples should be enough for explaining the grain refinement mechanism.
- Lines 207-209. You describe not the molten pool, but the section of rapidly solidified metal. They are very different, please, correct the explanation.
Thank you for pointing that out. We have replaced this incorrect description.
- Figure 14 is described very inaccurately. The 4 images contain abrasive grooves and delamination, but in different ratios. Fig. 4bd have especially intensive delamination. And, I disagree, that on Fig. 14d in the oval there are wear debris. It is a site of delamination. For more careful analyses, the other, higher magnification images should be made. All 5 alloys should be analyzed. The pores on Fig. 14d are close in shape and size to the Y2O3 “nanoparticles” from Figure 1b.
- Figure 13 should be discussed more broadly. Why the COF rapidly decreases, when Y2O3 content is 0.5%, and then increases? What is the reason? High COF is not a result of deep grooves. It is a reason, why the grooves become deeper. Please, find the actual reason of changing the COF. For this, as I recommend, use the higher magnification images (X2000, X5000).
Reply to comment 11 and 12: Thank you for your comment. Based on your suggestions, we used the higher magnification SEM images (X2000, X5000) to illustrate friction and wear mechanisms. For the S0, the worn surface showed deep grooves and severe plastic deformation (Figure 14a). The delamination and exfoliation phenomenon could be clearly observed. For the S1 and S2, small delamination and pits are observed on the smooth worn surface, suggesting a weak wear behavior in this case. Owing to their high hardness, a high-density of hard reinforcements dispersed in the soft matrix can effectively limit further material removal in the course of the wear process. However, the mean COF of the S4 increased to 0.65, and the resultant wear rate also increased slightly to (4.67 ± 0.18) ×10−4 mm3N-1m-1. The decrease in tribological performance in this instance was ascribed to the decrease in density and the grain coarsening (Figure 8b), which can easily split during sliding, resulting in the spalling of the worn surface and attendant limited wear resistance.
To further understand the friction and wear mechanism, the wear surface topography is shown in Figure 14. For the S0, the worn surface showed deep grooves and debris (Figure 14a). Therefore, a combined mechanism of abrasive and adhesive wear dominated, showing a relatively high average COF of ~0.709 and high wear rate of (5.59 ± 0.15) ×10−4 mm3N-1m-1 for S0. For the S1 and S2, small delamination and pits are observed on the smooth worn surface, revealing a weak wear behavior in this case and implying that the dominant wear mechanism changed to adhesion of a strain-hardened tribolayer. Owing to their high hardness, a high-density of hard reinforcements dispersed in the soft matrix can effectively limit further material removal in the course of the wear process. However, the COF of the S4 increased to 0.65, and the resultant wear rate also increased slightly to (4.67 ± 0.18) ×10−4 mm3N-1m-1. The decrease in tribological performance in this instance was ascribed to the decrease in density and the grain coarsening (Figure 8b), which can easily split during sliding, resulting in the spalling of the worn surface and attendant limited wear resistance.
- After the article is improved, please, rewrite the conclusions taking into account new results obtained.
Thank you for your comment, we have rewritten the conclusions.
- Adding 0.5 wt.% Y2O3 nanoparticles can significantly refine the grains from 1.742 to 0.873 μm. But further addition of Y2O3 nanoparticles will result in a grain size increase and the decrease of the relative density.
- The optimal nano-Y2O3 particle addition level is 0.5 wt.%, with a higher ultimate tensile strength of 500.3 MPa, a yield strength of 322.3 MPa, and an elongation of 9.7%. The Orowan strengthening effect and load-bearing strengthening effect both shows that the addition of nano- Y2O3 is beneficial to the grain refinement. However, since the grain size gradually increases and the relative density decreases as the addition level of Y2O3 passes 1 wt%, the strength of the material also experiences a decrease.
- The wear resistance of the Y2O3-AlSi10Mg nanocomposites is improved compared to that of the AlSi10Mg alloy. When adding 0.5 wt.% Y2O3 nanoparticles, the wear rate is about 39% lower than that of the AlSi10Mg alloys. But when the addition of Y2O3 increases, the wear performance gradually decreases.
At last, we want to thank the reviewer for all the helpful comments, which has definitely improved the quality of this work.
Yours sincerely,
Zhenyu

Reviewer 2 Report
In this manuscript, yttrium oxide nanoparticles with different weight percentages have been used to fabricate AlSi10Mg alloy composite by SLM method, and the mechanical and microstructural properties have been comprehensively investigated. The different sections of the paper are very well organized and the results are presented very well. Only minor modifications are needed in some sections.
The title of the papers should be improved: phrases such as comprehensive investigation, and mechanical properties must be added.
The novelty of the article should be clearly added to the abstract.
The introduction is very concisely written and this should be improved. The following sources can be used for this purpose. (Effect of welding thermal treatment on the microstructure and mechanical properties of nickel-based superalloy fabricated by selective laser melting, The high temperature flow behavior of additively manufactured Inconel 625 superalloy)
Lines 107-110: On what basis are the optimized parameters selected? It is also suggested to summarize these parameters in a table.
Figure 3 and its description (lines 11 to 113) should be moved to the Mechanical Properties section to complete this section.
The Phase composition and microstructural characterization section, especially optical evaluation, is completely a report of the results to which analysis should be added. It is recommended to use the following sources (Review of selective laser melting of magnesium alloys: advantages, microstructure and mechanical characterizations, defects, challenges, and applications, Microstructural origin and control mechanism of the mixed grain structure in Ni-based superalloys, Microstructure and mechanical properties of ultrasonic spot welding TiNi/Ti6Al4V dissimilar materials using pure Al coating,).
Author Response
Dear reviewer,
Thank you very much for your review comment. We have revied the manuscript according to your comments. Shown below is the detailed information on how I responded to the comments, highlighting the revisions made in the manuscript. When reading the information below, please note:
Text in italic style: the comments
Text in regular style: my responses to the comments.
Changes are highlighted in red color in the revised manuscript.
REVIEWER REPORT(S):
Comments to the Author
Recommendation: Publish after major revisions noted.
Comments: In this manuscript, yttrium oxide nanoparticles with different weight percentages have been used to fabricate AlSi10Mg alloy composite by SLM method, and the mechanical and microstructural properties have been comprehensively investigated. The different sections of the paper are very well organized and the results are presented very well. Only minor modifications are needed in some sections.
Questions:
- The title of the papers should be improved: phrases such as comprehensive investigation, and mechanical properties must be added.
Thank you for pointing that out, we have adjusted our title of the papers.
Microstructure and mechanical properties of nanoparticulate Y2O3 modified AlSi10Mg alloys manufactured by selective laser melting
- The novelty of the article should be clearly added to the abstract.
Thank you for your comment. We have adjusted the abstract especially the first line to better address the novelty and the urgent need for this study.
- The introduction is very concisely written and this should be improved. The following sources can be used for this purpose. (Effect of welding thermal treatment on the microstructure and mechanical properties of nickel-based superalloy fabricated by selective laser melting, The high temperature flow behavior of additively manufactured Inconel 625 superalloy).
Thank you for your comment. The articles you provided were very helpful to us and we have modified the introduction as you suggested.
- Lines 107-110: On what basis are the optimized parameters selected? It is also suggested to summarize these parameters in a table.
Thank you for pointing that out. The influence of laser power, scanning speed, hatching distance and layer thickness on the density of AlSi10Mg alloy was already studied in preliminary tests. The combination of process parameters with the highest density was selected as the optimized parameters for experimental research in this paper. However, obtaining such combination is not the focus of this paper, so we did not explain how it is obtained.
laser power(W) |
hatching distance(mm) |
scanning speed(mm/s) |
layer thickness(mm) |
200 |
0.12 |
1200 |
0.02 |
Table 2. The optimized SLM parameter combinations
- Figure 3 and its description (lines 110 to 113) should be moved to the Mechanical Properties section to complete this section.
Thank you for your comment. We have adjusted the position of Figure 3 and its description. We have also added a few lines describing the preparation of the coupons.
- The Phase composition and microstructural characterization section, especially optical evaluation, is completely a report of the results to which analysis should be added. It is recommended to use the following sources (Review of selective laser melting of magnesium alloys: advantages, microstructure and mechanical characterizations, defects, challenges, and applications, Microstructural origin and control mechanism of the mixed grain structure in Ni-based superalloys, Microstructure and mechanical properties of ultrasonic spot welding TiNi/Ti6Al4V dissimilar materials using pure Al coating,).
Thank you very much for your comment. We have added some analysis to the paper and have added the mentioned sources.
At last, we want to thank the reviewer for all the helpful comments, which has definitely improved the quality of this work.
Yours sincerely,
Zhenyu

Reviewer 3 Report
The paper is very interesting. Can be published as presented.
Author Response
We want to express our appreciation for your review comments.
Reviewer 4 Report
The paper documents the mechanical and tribology performance of AlSi10Mg with milled Y2O3 nanoparticle inclusions produced by L-PBF. An improvement in the microhardness and tensile strength was reported based on grain-refinement of alpha phase typically observed in the eutectic alloy. The results were compared to typical L-PBF processed AlSi10Mg. A thorough detailed analysis and characterisation of the microstructural development , defects introduced across varying grades of Y2O3 were reported. Some fundamental research practices need to be addressed and stated in the manuscript prior to acceptance, albeit these were minor details were not included. Otherwise, excellent overview for literature regarding additions of nano-particle inclusions in AlSi10Mg from a selection of authors. Thorough analysis of the micro-structure has been reported in this study. Overall, a recommendation to improve the manuscript would be highlighting the novelty or difference in the work using the Y203 nanoparticle inclusions has over existing studies in other work.
Additionally, why modifying existing AlSI10Mg. Are there not other interesting alloy combinations that could be explored?
General queries and feedback:
The only suspicion regarding the novelty of the paper is what is unique and specific benefit to the use of Y2O3 particles from the review of literature. It is not clear why this is different from other reinforcement particle systems used by other authors. This argument is very weak and needs improvement.
Few issues regarding the method of preparation:
The stated material composition of AlSi10Mg – was this measured by authors, or provided from the manufacturing specs? These can differ widely and need to be monitored, especially academic papers do not necessarily report this clearly or state the source of their measurement.
How was size of particle sizes determined for both powders? Nano-particles were not manufactured but supplied.
How were the nano-particles created – Figure 1b seems different order of scale to typical nano-particles?
No procedure for the ball milling and mixing. This should be stated so it is fully reproducible to the readers
A custom printer was used, therefore it would be good to have a photograph or schematic of this (or highlight in previous literature) because subtleties between printers exist and would be more convincing if the printing method was clear to the reader.
Tensile Coupon Sample Preparation
Not explicitly clear how many samples were prepared for each sample? This should be stated.
No indication how the samples were built and their orientation? Samples were machined, but were these near-net shape. These must be stated. A photograph of a built sample rather than post-process would remove further doubts into their preparation.
The density of S3, S4 reduces with increased content of reinforcement particles. These defects. This is not touch upon greatly. It is appreciated laser parameters were not changed but 98% dense parts for some users is not acceptable for density. Could this impact the reduction in strength observed? It is briefly touched upon in Page 11.
After reading the paper, it is still unclear why S1 is the best performer in terms of mechanical strength and wear performance. This could be better highlighted in the conclusions.
Specific comments:
Title:
Generally L-PBF is now accepted as the correct terminology for SLM.
Abstract:
1st sentence is useless and should be removed.
Strange unit combination for wear rate: mm3N-1m-1
Introduction
Revise the first paragraph.
The first 5 lines are really poor and are general statements on metal AM. Cite individual aspects of the benefits to SLM, rather than clustering them at the end of sentence.
Methodology:
· Formatting: Table 1 is split across two pages
· Scale bar on Figure 2a – it seems to differ from that in Figure 1a – was this particle at the lower-end of the particle size distribution
· Equation (1,2) formatted strangely (stretched)
· (Pg3 Line 107)Typically Argon concentration is specified in PPM
· (Pg3 Line 112) “dog-bone-shaped plates” could be expressed as flat tensile coupons/bars
· Figure 9: Expand the figure to include the sample type so it is easier to infer which graph refers to the concentration of Y203
Bibliography:
Reference 6, 8 - Literature is over 8 years old. Consider more recent sources of general literature regarding SLM and AlSi10Mg.
Author Response
Dear reviewer,
Thank you very much for your review comment. We have revied the manuscript according to your comments. Shown below is the detailed information on how I responded to the comments, highlighting the revisions made in the manuscript. When reading the information below, please note:
Text in italic style: the comments
Text in regular style: my responses to the comments.
Changes are highlighted in red color in the revised manuscript.
REVIEWER REPORT(S):
Comments to the Author
Recommendation: Publish after major revisions noted.
Comments: The paper documents the mechanical and tribology performance of AlSi10Mg with milled Y2O3 nanoparticle inclusions produced by L-PBF. An improvement in the microhardness and tensile strength was reported based on grain-refinement of alpha phase typically observed in the eutectic alloy. The results were compared to typical L-PBF processed AlSi10Mg. A thorough detailed analysis and characterisation of the microstructural development , defects introduced across varying grades of Y2O3 were reported. Some fundamental research practices need to be addressed and stated in the manuscript prior to acceptance, albeit these were minor details were not included. Otherwise, excellent overview for literature regarding additions of nano-particle inclusions in AlSi10Mg from a selection of authors. Thorough analysis of the micro-structure has been reported in this study. Overall, a recommendation to improve the manuscript would be highlighting the novelty or difference in the work using the Y203 nanoparticle inclusions has over existing studies in other work. Additionally, why modifying existing AlSI10Mg. Are there not other interesting alloy combinations that could be explored? The title and the intentions declared in the abstract correspond to the contents of the paper. Some of the references could be more related to the subject of the paper.
Questions:
- The only suspicion regarding the novelty of the paper is what is unique and specific benefit to the use of Y2O3 particles from the review of literature. It is not clear why this is different from other reinforcement particle systems used by other authors. This argument is very weak and needs improvement.
Thank you for pointing that out. We are sorry for not addressing the novelty enough. We have rewritten the introduction section concerning the novelty. Our team has done plenty of works in exploring the unique properties of different rare earth oxides, in polishing and in alloy modification. Oxide dispersion strengthening is a hot topic and the use of rare earth oxide in alloy modification is already proven effective. In this study, we have discovered that Y2O3 is not only a perfect candidate for modifying Si in Al-Si alloys, it is also a fine match for Al which can help refine grain according to the E2EM theory. Therefore, the mechanism behind grain refinement of the Y2O3 modified AlSi10Mg is very complex and of great importance.
Rare earth oxides like Y2O3 are considered perfect chemical modifiers for Al-Si alloys, since the ratio of atomic radius of Y relative to that of Si is very close to 1.646. The mechanism of silicon modification in aluminum–silicon alloys: Impurity induced twinning [23]. In Moussa’s study, it was discovered that the addition of small amounts of Y2O3 nanoparticles to Al-Si alloys can reduce the grain size of α-Al and enhance the strength and plasticity of cast samples [24]. Utilizing the refinement and modification mechanism related to twinning of {111}, the fracture mode gradually transfers from cleavage transgranular fracture to the ductile-brittle fracture mode. Moreover, Y2O3 was applied to improve the properties of 316L [25], pure tungsten [26], Inconel 625[27], and other materials processed by SLM, further proving its effectiveness as an oxide dispersion strengthening addition. However, the effect of Y2O3 nanoparticles on the organization and mechanical properties of AlSi10Mg alloys has not been explored yet. Whether the same grain refinement effect can be achieved for AlSi10Mg is yet unknown. Therefore, it is important to investigate the enhancement capability and the strengthening mechanism of Y2O3 nanoparticles for the AlSi10Mg alloys.
- The stated material composition of AlSi10Mg – was this measured by authors, or provided from the manufacturing specs? These can differ widely and need to be monitored, especially academic papers do not necessarily report this clearly or state the source of their measurement.
Thank you for your comment. The stated material composition of AlSi10Mg was provided by the manufacturer. Indeed, the manufacturing specs and the measured composition can vary a lot in many cases. For reproducibility and stability of our study, we have chosen a manufacturer that is well known in the industry (in China mainland) and here are a few works that uses printing powders from the same manufacturer. We believed that the composition provided by the manufacturer is enough for this study since the main focus was to study the grain refinement mechanism of Y2O3.
Here are some researches that use material provided by Jiangsu Vilory Advanced Material:
[1] Yang, Tianhai, et al. "Preparation of nanostructured CoCrFeMnNi high entropy alloy by hot pressing sintering gas atomized powders." Micron 147 (2021): 103082.
[2] Xiong, Ke, et al. "Cooling-Rate Effect on Microstructure and Mechanical Properties of Al0. 5CoCrFeNi High-Entropy Alloy." Metals 12.8 (2022): 1254.
[3] Ge, Jinguo, et al. "Effect of volume energy density on selective laser melting NiTi shape memory alloys: microstructural evolution, mechanical and functional properties." Journal of Materials Research and Technology 20 (2022): 2872-2888.
[4] Zhang, Yubei, et al. "Digital light processing 3D printing of AlSi10Mg powder modified by surface coating." Additive Manufacturing 39 (2021): 101897.
and more…
- How was size of particle sizes determined for both powders? Nano-particles were not manufactured but supplied.
We are sorry for causing the confusion. A gas-atomized AlSi10Mg powder is supplied by Jiangsu Vilory Advanced Material Technology Co., Ltd, China and the commercial nano-Y2O3 is supplied by Beijing Zhongke Yannuo New Material Technology Co., Ltd, China. Thesize of particles is provided by the company. The particle size was roughly measured under SEM each time we purchased a new batch. But we considered that a small deviation of the particle size is of small importance in this work. Moreover, before each experiment, we have sieved out AlSi10Mg particles larger than 60 μm.
- How were the nano-particles created – Figure 1b seems different order of scale to typical nano-particles?
Thank you for pointing that out, our Figure may mislead readers. We have already made the replacement. Nanoparticles are very prone to agglomeration, and nanoscale particles can be observed through higher magnification scanning electron microscopy photos. Through ball milling, the nanoparticles can be evenly distributed with AlSi10Mg particles without affecting subsequent experiments.
- No procedure for the ball milling and mixing. This should be stated so it is fully reproducible to the readers.
Thank you for your comment. We have added it to the paper.
The powder mixture was sealed in stainless steel bowls with a ball-to-powder ratio of 3:1. The rotation rate was set at 250 rpm with a total milling time of 4 h. An interval of 10 min was set after each 20 min of milling in order to avoid overheating of powder.
- A custom printer was used, therefore it would be good to have a photograph or schematic of this (or highlight in previous literature) because subtleties between printers exist and would be more convincing if the printing method was clear to the reader.
Thank you for pointing that out. The SLM printer used in the paper is a customed build of our partners' equipment. As shown in Figure (a), the CR-SLM100 is manufactured by Shandong CharmRay laser technology co. LTD and is a commercial model. Figure (b) shows the equipment we used in our experiment, and we have added Figure (b) to the paper. We believed that the framework of a commercial model is good enough for our test and indeed the printing stability was very good throughout all our tests.
- Not explicitly clear how many samples were prepared for each sample? This should be stated.
Thank you very much for pointing out that. We have added this to the paper.
Three tests were performed on each sample, and the average value was taken as the final results to ensure their mechanical reliability.
Please refer to the revised manuscript for all the rewritten lines, which are marked in red.
- No indication how the samples were built and their orientation? Samples were machined, but were these near-net shape. These must be stated. A photograph of a built sample rather than post-process would remove further doubts into their preparation.
Thank you for pointing that out. We have added a picture of the prepared samples to the paper. All specimens are removed from the substrate by Wire cut Electrical Discharge Machining. Before mechanical tests, all specimens were pre-treated using mechanically ground and polished using standard metallographic methods.
- The density of S3, S4 reduces with increased content of reinforcement particles. These defects. This is not touch upon greatly. It is appreciated laser parameters were not changed but 98% dense parts for some users is not acceptable for density. Could this impact the reduction in strength observed? It is briefly touched upon in Page 11.
Thank you for your comment. The tensile, friction and wear properties of composite materials were mainly related to the relative density, hardness, microstructure and distribution of reinforcing particles (Selective laser melting of TiN nanoparticle-reinforced AlSi10Mg composite: Microstructural, interfacial, and mechanical properties, Journal of Materials Processing Technology, https://doi.org/10.1016/j.jmatprotec.2020.116618.).
The density of S3, S4 reduces with increased content of reinforcement particles, and that lead to the reduction in strength. We have added a few lines to better addressed the mechanism.
- After reading the paper, it is still unclear why S1 is the best performer in terms of mechanical strength and wear performance. This could be better highlighted in the conclusions. Please correct the titles of the figures and also of the chapter 3.3 (start with capitals).
We have modified the conclusion, corrected the mistakes and have check the whole manuscript for similar problems.
- Generally L-PBF is now accepted as the correct terminology for SLM.
Thank you, but we would like to keep calling it SLM for now for consistency with other works from our team.
- 1st sentence is useless and should be removed.
Thank you for pointing that out, we have rewritten the first few lines of the abstract.
- Strange unit combination for wear rate: mm3N-1m-1.
Thank you for your comment. Actually, it is a widely accepted unit, for mm3 is the worn volume, N the applied force and m the total worn distance. In combination, it means X volume of material was worn under a specific load and after travelling a specific distance.
- The first 5 lines are really poor and are general statements on metal AM. Cite individual aspects of the benefits to SLM, rather than clustering them at the end of sentence.
Thank you for your comment, we have rewritten the first paragraph.
Additive manufacturing is a potentially disruptive technology across multiple industries, including automotive industries, biomedical and aerospace [1]. The principle of additive manufacturing is to accumulate materials point by point on the surfaces and solidify them into bodies layer by layer [2], enabling the formation of complex parts [3] and increasing product customization [4]. As a dominant AM technology, SLM is highly appreciated for its relatively low surface roughness, high geometric accuracy, and good mechanical properties [5,6]. Based on previously established CAD data, SLM selectively melts a fixed-thickness powder layer on the solidified powder bed, and quickly, layer by layer produces three-dimensional parts with complex shapes [7,8].
- Formatting: Table 1 is split across two pages.
Thank you for pointing that out. We have adjusted the layout of the table.
- Scale bar on Figure 2a – it seems to differ from that in Figure 1a – was this particle at the lower-end of the particle size distribution.
Thank you for your comment. This particle was at the lower-end of the particle size distribution. Our main objective was to find a complete particle stating that Y2O3 nanoparticles were uniformly distributed on the surface of AlSi10Mg particle, and its size had no effect on subsequent experiments.
17.Equation (1,2) formatted strangely (stretched).
We have adjusted the Equation (1,2).
- (Pg3 Line 107)Typically Argon concentration is specified in PPM.
Thank you for pointing that out. We have added more information for the atmosphere in the paper.
- (Pg3 Line 112) “dog-bone-shaped plates” could be expressed as flat tensile coupons/bars.
We have changed“dog-bone-shaped plates” to flat tensile coupons.
20.Figure 9: Expand the figure to include the sample type so it is easier to infer which graph refers to the concentration of Y203
Thank you for pointing that out. We have adjusted Figure 9.
- Reference 6, 8 - Literature is over 8 years old. Consider more recent sources of general literature regarding SLM and AlSi10Mg.
Thank you for your comment. We have replaced references 6,8 with more recent articles.
At last, we want to thank the reviewer for all the helpful comments, which has definitely improved the quality of this work.
Yours sincerely,
Zhenyu

Reviewer 5 Report
The work presents the microstructural evaluation of yttrium oxide infused AlSi10Mg alloys through selective laser melting process. I found the paper to be well-organised, properties well-described and results nicely presented. However I have some minor comments as follows:
1. The error bars in each of the plots are barely visible and needs an update.
2. I found some of the publications to be old and obsolete. The authors are advised to include references from last 5 years.
3. Can the authors describe the type of wear in the samples? What type of wear is it? Please elaborate in the text.
4. Conclusions need the overall results and not explanation to why the results have come up.
Author Response
Dear reviewer,
Thank you very much for your review comment. We have revied the manuscript according to your comments. Shown below is the detailed information on how I responded to the comments, highlighting the revisions made in the manuscript. When reading the information below, please note:
Text in italic style: the comments
Text in regular style: my responses to the comments.
Changes are highlighted in red color in the revised manuscript.
REVIEWER REPORT(S):
Comments to the Author
Recommendation: Publish after major revisions noted.
Comments: The work presents the microstructural evaluation of yttrium oxide infused AlSi10Mg alloys through selective laser melting process. I found the paper to be well-organised, properties well-described and results nicely presented. However I have some minor comments as follows:
Questions:
- The error bars in each of the plots are barely visible and needs an update.
Thank you for pointing that out, we have adjusted the error bars.
- I found some of the publications to be old and obsolete. The authors are advised to include references from last 5 years.
Thank you for your comment. We have added a few more references that are from the last 5 years. As for the old references, we decided to keep it since they were the first to report similar mechanism.
- Can the authors describe the type of wear in the samples? What type of wear is it? Please elaborate in the text.
Thank you, we have rewritten this part of the friction and wear mechanism.
To further understand the friction and wear mechanism, the wear surface topography is shown in Figure 14. For the S0, the worn surface showed deep grooves and debris (Figure 14a). Therefore, a combined mechanism of abrasive and adhesive wear dominated, showing a relatively high average COF of ~0.709 and high wear rate of (5.59 ± 0.15) ×10−4 mm3N-1m-1 for S0. For the S1 and S2, small delamination and pits are observed on the smooth worn surface, revealing a weak wear behavior in this case and implying that the dominant wear mechanism changed to adhesion of a strain-hardened tribolayer. Owing to their high hardness, a high-density of hard reinforcements dispersed in the soft matrix can effectively limit further material removal in the course of the wear process. However, the COF of the S4 increased to 0.65, and the resultant wear rate also increased slightly to (4.67 ± 0.18) ×10−4 mm3N-1m-1. The decrease in tribological performance in this instance was ascribed to the decrease in density and the grain coarsening (Figure 8b), which can easily split during sliding, resulting in the spalling of the worn surface and attendant limited wear resistance.
- Conclusions need the overall results and not explanation to why the results have come up. Thank you for pointing that out. we have rewritten the conclusions.
- Adding 0.5 wt.% Y2O3 nanoparticles can significantly refine the grains from 1.742 to 0.873 μm. But further addition of Y2O3 nanoparticles will result in a grain size increase and the decrease of the relative density.
- The optimal nano-Y2O3 particle addition level is 0.5 wt.%, with a higher ultimate tensile strength of 500.3 MPa, a yield strength of 322.3 MPa, and an elongation of 9.7%. The Orowan strengthening effect and load-bearing strengthening effect both shows that the addition of nano- Y2O3 is beneficial to the grain refinement. However, since the grain size gradually increases and the relative density decreases as the addition level of Y2O3 passes 1 wt%, the strength of the material also experience a decrease.
- The wear resistance of the Y2O3-AlSi10Mg nanocomposites is improved compared to that of the AlSi10Mg alloy. When adding 0.5 wt.% Y2O3 nanoparticles, the wear rate is about 39% lower than that of the AlSi10Mg alloys. But when the addition of Y2O3 increases, the wear performance gradually decreases.
At last, we want to thank the reviewer for all the helpful comments, which has definitely improved the quality of this work.
Yours sincerely,
Zhenyu
Round 2
Reviewer 1 Report
Dear Authors!
You did a good job to improve the manuscript, but a few minor corrections are still required.
Line 119. ...atmosphere of less than 0.1 MPa.... It is not very clear. Would You please rewrite it like "at argon pressure P<0.1 MPa
Line 144. In Figure 3b, the printing direction is not specified.
Figure 7. Please, correct the word "Celluar". It should be "Cellular".
Author Response
Dear reviewer,
Thank you very much for your review comment. We have revied the manuscript according to your comments. Shown below is the detailed information on how I responded to the comments, highlighting the revisions made in the manuscript. When reading the information below, please note:
Text in italic style: the comments
Text in regular style: my responses to the comments.
Changes are highlighted in red color in the revised manuscript.
REVIEWER REPORT(S):
Comments to the Author
Recommendation: Publish after major revisions noted.
Comments: You did a good job to improve the manuscript, but a few minor corrections are still required.
Questions:
- Line 119. ...atmosphere of less than 0.1 MPa.... It is not very clear. Would You please rewrite it like "at argon pressure P<0.1 MPa
Thank you for pointing that out, we have already rewritten it.
- Line 144. In Figure 3b, the printing direction is not specified.
Thank you for your comment. But we are not sure what “printing direction” here refers to. If it means the building direction, we have added a mark on Figure 3b for that. If it means scanning direction of the laser, we considered that the sentences in the manuscript are enough for describing how the printing process was.
- Figure 7. Please, correct the word "Celluar". It should be "Cellular".
Thank you for your comment. We have corrected this mistake.
At last, we want to thank the reviewer for all the helpful comments, which has definitely improved the quality of this work.
Yours sincerely,
Zhenyu